# CONTRASTIVE CODE REPRESENTATION LEARNING

## ABSTRACT

Machine-aided programming tools such as automated type predictors and autocomplete are increasingly learning-based. However, current approaches predominantly rely on supervised learning with task-specific datasets. We propose *Contrastive Code Representation Learning* (ContraCode), a self-supervised algorithm for learning task-agnostic semantic representations of programs via contrastive learning. Our approach uses no human-provided labels, only the raw text of programs. ContraCode optimizes for a representation that is invariant to semantic-preserving code transformations. We develop an automated source-to-source compiler that generates textually divergent variants of source programs. We then train a neural network to identify variants of anchor programs within a large batch of non-equivalent negatives. To solve this task, the network must extract features representing the functionality, not form, of the program. In experiments, we pre-train ContraCode with 1.8M unannotated JavaScript methods mined from GitHub, then transfer to downstream tasks by fine-tuning. Pre-training with ContraCode consistently improves the F1 score of code summarization baselines and top-1 accuracy of type inference baselines by 2% to 13%. ContraCode achieves 9% higher top-1 accuracy than the current state-of-the-art static type analyzer for TypeScript. Finally, representations learned through a hybrid contrastive and reconstruction objective transfer in zero-shot to code clone detection with +10% AUROC over a static text similarity measure and +5% over reconstruction alone.

## 1 INTRODUCTION

Programmers increasingly rely on machine-aided programming tools to aid software development (Kim et al., 2012). However, the wide diversity of programs encountered in practice limits the generalization of hand-written rules. Catching semantic bugs such as naming errors requires deeper language understanding, motivating learning-based programming tools. Recent work uses machine learning for bug detection (Pradel & Sen, 2018) and optimization (Mendis et al., 2019). Consider predicting the type of the variable declaration "var median = ...;". Static analysis fails as the type is underspecified, but the variable name indicates the statement is a float.

Programming language datasets suffer from scarce annotations due to the time and expertise required to label. State-of-the-art approaches generally rely on either (1) synthetic supervised datasets or (2) self-supervised pre-training. Synthetic auto-generated labels have been used for method naming (Alon et al., 2019a;b) and bug detection (Ferenc et al., 2018; Benton et al., 2019; Pradel & Sen, 2018). However, synthetic code datasets suffer from duplication issues (Allamanis, 2019) and biases (Shin et al., 2019) which degrade generalization. Moreover, auto-generated data does not cover the diverse program behaviors encountered in the wild.

In contrast, self-supervised learning can leverage large open-source repositories such as GitHub with limited or no annotations. Inspired by the success of pre-training in natural language processing, recent work uses self-supervision to learn code representations. Authors have explored context-based token embeddings (Ben-Nun et al., 2018) and masked language modeling, where tokens are corrupted and reconstructed (Feng et al., 2020; Kanade et al., 2020) However, reconstruction focuses on superficial language reasoning and does not explicitly address the underlying program functionality. The resulting models attend to program implementation specifics such as variable names.

We hypothesize that *programs with the same functionality should have the same underlying representation* for downstream code understanding tasks, a principle illustrated in Fig. 1. While it is time

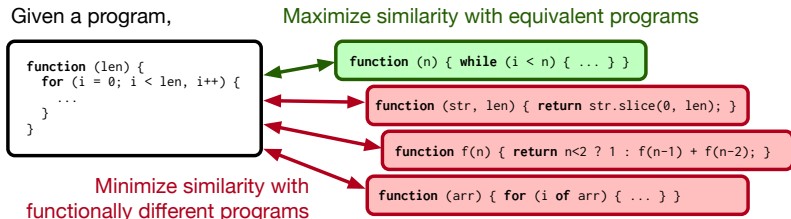

Figure 1: Programs with the same functionality should have the same underlying representation. ContraCode learns such representations with contrastive learning: the network is trained to find equivalent programs among many distractors, encoding semantics into the representation.

intensive to identify equivalent programs in a large corpus, it is cheap to leverage static compiler transformations to automatically generate many equivalent versions of a particular source program.

In this work, we develop ContraCode, a self-supervised representation learning algorithm that uses source-to-source compiler transformation techniques (e.g., dead code elimination, obfuscation and constant folding) to generate syntactically diverse but functionally equivalent programs. ContraCode uses these equivalent programs to construct a challenging *discriminative* pretext task that requires the model to identify equivalent programs out of a large dataset of distractors. In doing so, it has to embed the functionality, not the form, of the code. In essence, the domain knowledge from our code transformations induces the knowledge of the structure of programs onto learned representations. The contributions of our work include:

1. the novel use of compiler-inspired transformations as data augmentations for code,
2. the concept of program representation learning based on functional equivalence, and
3. a detailed analysis of architectures, code transforms and pre-train strategies, where ContraCode improves static type inference top-1 accuracy by 9%, learned inference by 2% – 13%, summarization F1 score by up to 8% and clone detection AUROC by 5% – 10%.

## 2    RELATED WORK

**Self-supervised learning** (SSL) is a general representation learning strategy where some dimensions or attributes of a datapoint are predicted from the remaining parts. These methods are unsupervised in the sense that they do not rely on labels, but SSL tasks often adapt losses and architectures designed for supervised learning. Self-supervised pre-training has yielded large improvements in both NLP (Howard & Ruder, 2018; Devlin et al., 2018; Radford et al., 2018; 2019) and computer vision (Mahajan et al., 2018) by improving generalization (Erhan et al., 2010; Hao et al., 2019). Weak visual features, such as orientation (Gidaris et al., 2018), color (Zhang et al., 2016), and context (Pathak et al., 2016), are meaningful signals for representations (Mahajan et al., 2018).

**Contrastive learning** unifies many past SSL approaches that compare pairs or collections of similar and dissimilar items (Hadsell et al., 2006). Rather than training the network to predict labels or reconstruct data, contrastive methods minimize the distance between the representations of similar examples (positives) while maximizing the distance between dissimilar examples (negatives). Examples include Siamese networks (Bromley et al., 1994) and triplet losses (Schroff et al., 2015). Contrastive predictive coding (Oord et al., 2018; Hénaff et al., 2019) learns to encode chunks of sequential data to predict of future chunks with the InfoNCE loss, a variational lower bound on mutual information between views of the data (Tian et al., 2019; Wu et al., 2020) inspired by noise-constrastive estimation (Gutmann & Hyvärinen, 2010). In instance discrimination tasks (Wu et al., 2018), views and not pieces of an entire image are compared. SimCLR (Chen et al., 2020a) and Momentum Contrast (He et al., 2019; Chen et al., 2020b) recently made progress by using many negatives for dense loss signal. Beyond images, InfoNCE has been applied to NLP (Chuang et al., 2020; Giorgi et al., 2020), but may require supervision (Fang & Xie, 2020).

**Code representation learning**    There has been substantial work on architectures and tasks for machine learning on code (Allamanis et al., 2018). We adopt the summarization task of Alon et al.

```
function x(maxLine) {
  const section = {
    text: '',
    data
  };

  for (; i < maxLine; i += 1) {
    section.text += `${lines[i]}\n`;
  }

  if (section) {
    parsingCtx.sections.push(section);
  }
}
```
Original JavaScript method

```
function x(t) {
  const n = {
    'text': '',
    'data': data
  };
  for (;i < t; i += 1) {
    n.text += lines[i] + '\n';
  }
  n && parsingCtx.sections.push(n);
}
```
Renamed variables, explicit object style,
explicit concatenation, inline conditional

```
function x(t){const
n={'text':'','data':data};for(;i<t;i+=
1)n.text+=lines[i]
+'\n';n&&parsingCtx.sections.push(n)}
```
Mangled source with
compressed whitespace

Figure 2: A JavaScript method from the unlabeled training set with two automatically generated semantically-equivalent programs. The original method is from the StackEdit Markdown editor.

(2019a), and the variable type inference task of DeepTyper (Hellendoorn et al., 2018). Other authors have explored summarization (Movshovitz-Attias & Cohen, 2013; Allamanis et al., 2016; Iyer et al., 2016) and type inference (Pradel et al., 2019; Pandi et al., 2020; Wei et al., 2020; Allamanis et al., 2020; Bielik & Vechev, 2020) with different languages and datasets. The tree or graph structure of code can be exploited to encode invariances in the representation. Inst2vec (Ben-Nun et al., 2018) locally embeds individual statements in LLVM IR by processing a contextual flow graph with a context prediction objective (Mikolov et al., 2013). Tree-Based CNN embeds the Abstract Syntax Tree (AST) nodes of high-level source code. Code2seq (Alon et al., 2019a) embeds AST paths with an attention-based encoder and LSTM decoder for supervised sequence-to-sequence tasks. Kanade et al. (2020); Feng et al. (2020) pre-train the Transformer (Vaswani et al., 2017) on code using the masked language modeling objective (Devlin et al., 2018), an instance of the cloze task (Taylor, 1953) where the model reconstructs corrupted tokens. Recurrent networks have also been pre-trained on code (Hussain et al., 2020) as language models (Peters et al., 2018; Karampatsis & Sutton, 2020). Wang & Christodorescu (2019); Wang & Su (2019) assess the stability of program analyzers under semi-automated program transformations. Concurrent work by Rabin & Alipour (2020) found that code2vec and code2seq often change their classifications when statements are permuted, variables are renamed, or other-semantic preserving transformations are applied.

## 3 Method: Contrastive Code Representation Learning

Understanding program functionality and global structure is important for difficult tasks like summarizing code in natural language. For these problems, learned code representations should be similar for functionally equivalent programs and dissimilar for non-equivalent programs (Figure 1). The principle of contrastive learning offers a simple objective for learning such representations if data can be organized into pairs of *positives* and *negatives*. We use each pair to shape representation space, drawing positives together and pushing negatives apart. However, a major question remains: *given an unlabeled corpus of programs, how do we identify or generate similar programs?* We address this question in Sec. 3.1, then introduce our learning framework in Sec. 3.2.

### 3.1 Compilation as Data Augmentation

Modern programming languages afford great flexibility to software developers, allowing them to implement the same desired functionality in different ways. Crowdsourced datasets mined from developers, such as GitHub repositories, have many near-duplicates in terms of textual similarity (Allamanis, 2019), and are bound to contain even more functional equivalences for common tasks. Satisfiability solvers can identify these equivalent programs (Joshi et al., 2002; Bansal & Aiken, 2006), but functional equivalence is also undecidable in general (Rice, 1953). Also, formal documentation of semantics is required. Programs can instead be compared approximately using test-cases (Massalin, 1987), but this is costly and requires executing untrusted code.

Instead of *searching for equivalences*, we propose *correct by construction data augmentation*. Our insight is to apply source-to-source compiler transformations to unlabeled code to generate many variants with the same functionality. For example, dead-code elimination (DCE) is a common compiler optimization that removes operations that leave the output of a function unchanged. While

| Code compression | | Identifier modification | |
|---|---|---|---|
| ✓ | Reformatting (R) | ✓ | Variable renaming (VR) |
| ✓ | Beautification (B) | ✓ | Identifier mangling (IM) |
| ✓ | Compression (C) | | **Regularization** |
| ✓ | Dead-code elimination (DCE) | ✓ | Dead-code insertion (DCI) |
| ✓ | Type upconversion (T) | ✓ | Subword regularization (SW) |
| ✓ | Constant folding (CF) | ✗ | Line subsampling (LS) |

✓ = semantics-preserving transformation   ✗ = lossy transformation

Table 1: We augment programs with 11 automated source-to-source compiler transformations. 10 of the 11 transformations are correct-by-construction and do not modify operational semantics. More details are in Section A.3.

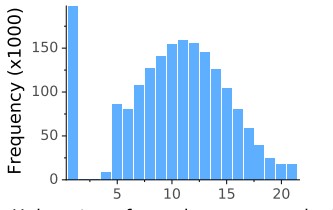

Figure 3: Histogram of the number of transformed variants per method during pre-training.

DCE preserves program functionality, Wang & Christodorescu (2019) find that up to 12.7% of the predictions of current algorithm classification models change after DCE—supervised datasets were not enough to acquire the domain knowledge that DCE does not matter.

A particular source code sequence, e.g. "W*x + b" is parsed unambiguously into a tree-structured representation "(+ (* W x) b)". This tree is then transformed by automated traversal algorithms. A rich body of prior programming language work explores parsing then tranforming Abstract Syntax Trees to optimize a program prior to machine code generation. If source code is output rather than machine code, this is called source-to-source transformation. Source-to-source transformations are common for optimization and obfuscation purposes in dynamic languages like JavaScript. If each transformation preserves code functionality, then any composition also preserves code functionality.

We leverage the Babel and Terser compiler infrastructure tools for JavaScript (McKenzie et al., 2020; Santos et al., 2020) to parse code into an Abstract Syntax Tree (AST) and then perform correctness-preserving transformations on method bodies. Table 1 and Appendix A.3 list all transformations, but we broadly group program transformations into three categories. **Code compression** changes the syntactic structure of code and performs correct-by-construction transformations such as pre-computing constant expressions at compile time. **Identifier modification** substitutes method and variable names with random tokens, thereby masking part of the semantic information in programs. Finally, transformations for **Regularization** improve model generalization by reducing the number of trivial positive pairs with high text overlap; this group potentially modifies program semantics through the line subsampling pass.

### 3.2 CONTRASTIVE PRE-TRAINING

Representations of semantically equivalent programs (positives) should have representations that each are closer to each other than semantically dissimilar programs (negatives). Contrastive learning is a natural framework to induce invariances into a model by attracting positives while repelling negatives. To adapt recent contrastive learning objectives for images to code representation learning, we leverage the augmentations discussed in Section 3.1.

We extend the Momentum Contrast method (He et al., 2019) that was designed for image representation learning. Our training procedure is depicted in Figure 4. Each transformation is a function $\tau : \mathcal{P} \to \mathcal{P}$, where the space of programs $\mathcal{P}$ is composed of both the set of valid ASTs and the set of programs in source form. At the beginning of an iteration, a batch of programs is sampled from a large database. Each program $x$ in the batch is transformed twice using two different, random subsets of transformations to derive textually different *query programs* and *key programs* according to Algorithm 1. Unlike computer vision data augmentations such as random cropping that are stochastic, our compiler-based transformations are deterministic.

To produce a diverse set of transformed programs, we randomly apply a subset of available compiler passes in a pre-specified order, applying transform $\tau_i$ with probability $p_i$. Intermediate programs are converted between AST and source form as needed. As all augmentations are precomputed, we deduplicate programs variants before pre-training. Figure 3 measures this diversity. 89% of the JavaScript functions in our dataset have more than one alternative after applying 20 random sequences of transformations. The remaining programs without syntactically distinct alternatives

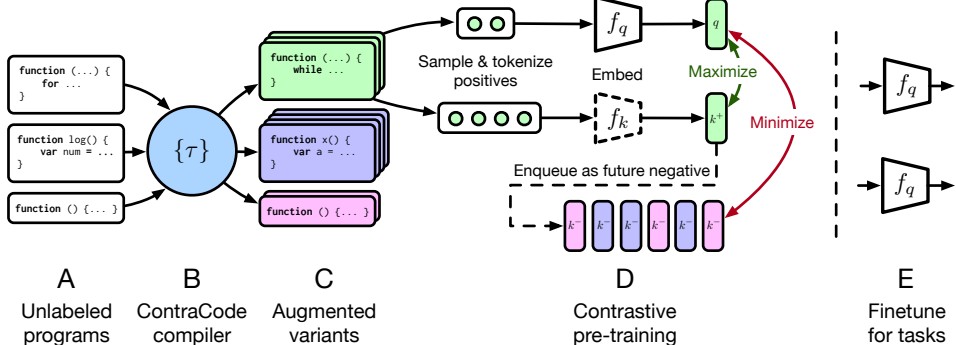

Figure 4: ContraCode pre-trains a neural program encoder $f_q$ and transfers it to downstream tasks. **A-B.** Unlabeled programs are transformed **C.** into augmented variants. **D.** We pre-train $f_q$ by maximizing similarity of embeddings of *positive* program pairs–variants of the same program–and minimizing similarity with queue of cached negatives. **E.** $f_q$ is fine-tuned on smaller labeled datasets.

include one-line functions that are obfuscated. We apply subword regularization (Kudo, 2018) as a final transformation to derive different tokenizations every batch, so pairs will still differ. All transformations are fast; our compiler transforms 300 functions per second on a single CPU core.

To reduce memory consumption during pre-training, we enqueue past batches to cache activations for negative samples. These cached samples are valid negatives if the queue is smaller than the dataset size. Following He et al. (2019), the query encoder $f_q$ is trained via gradient descent while the key encoder $f_k$ is trained slowly via an exponential moving average (EMA) of the query encoder parameters. The EMA update stabilizes the pre-computed key embeddings across training iterations. Since keys are only embedded once per epoch, we use a very large set of negatives, over 100K, with minimal additional computational cost and no explicit hard negative mining.

ContraCode supports different encoder architectures. We evaluate contrastive pre-training of Transformer (Vaswani et al., 2017) and BiLSTM (Schuster & Paliwal, 1997; Huang et al., 2015) architectures, with specific details in Section 4.

**Pre-training objective** The contrastive objective maximizes the similarity of positives without collapsing onto a single representation. Like He et al. (2019), we use InfoNCE (Oord et al., 2018), a tractable objective that frames contrastive learning as a classification task: can the positives be identified among a batch of sampled negatives? InfoNCE computes the probability of classifying the positive (transformed program) by taking the softmax of representation similarities across a batch of negatives. Equation (1) shows the InfoNCE loss for instance discrimination from He et al. (2019), a function whose value is low when $q$ is similar to the positive key embedding $k^+$ and dissimilar to negative key embeddings $k^-$. $t$ is a temperature hyperparameter proposed by Wu et al. (2018).

$$\mathcal{L}_{q,k^+,k^-} = -\log \frac{\exp(q \cdot k^+/t)}{\exp(q \cdot k^+/t) + \sum_{k^-} \exp(q \cdot k^-/t)} \tag{1}$$

The query representation $q = f_q(x^q)$ is computed by the encoder network $f_q$, and $x^q$ is a query program. Likewise, $k = f_k(x^k)$ using the EMA key encoder $f_k$. Views $x^q, x^k$ depend on the specific domain and pretext task. In our case, the views are tokenized representations of the augmented programs, and the summation $\sum_{k^-}$ in the normalizing denominator is taken over the queue of pre-computed negatives as well as other non-matching keys in the batch.

**Transfer learning** After pre-training converges, the encoder $f_q$ is transferred to downstream tasks. As the output space of the task can differ from the encoder, we add a task-specific MLP or Transformer decoder after $f_q$, then train the resulting network end-to-end on task data.

## 4 EXPERIMENTS

We evaluate whether self-supervised pre-training with ContraCode improves JavaScript and TypeScript code analysis. We benchmark on (1) extreme code summarization (Allamanis et al., 2016)

---

**Algorithm 1** Stochastic augmentation of programs with two possible encodings (AST or source).

---

1: **Input:** Program source $x$, transformation functions $\tau_1, \ldots \tau_k$, transform probabilities $p_1, \ldots p_k$
2: $\mathcal{V} \leftarrow \{x\}$, a set of augmented program variants
3: **for** SAMPLE $i \leftarrow 1 \ldots N$ **do**
4:    $x' \leftarrow x$
5:    **for** transform $t \leftarrow 1 \ldots k$ **do**
6:       Sample $y_t \sim \text{Bernoulli}(p_t)$
7:       **if** $y_t = 1$ **then**
8:          **if** REQUIRESAST($\tau_t(\cdot)$) and $\neg$ISAST($x'$) **then** $x' \leftarrow$ PARSETOAST($x'$)
9:          **else if** $\neg$REQUIRESAST($\tau_t(\cdot)$) and ISAST($x'$) **then** $x' \leftarrow$ LOWERTOSOURCE($x'$)
10:         $x' \leftarrow \tau_t(x')$
11:       **end if**
12:    **end for**
13:    **if** ISAST($x'$) **then** $x' \leftarrow$ LOWERTOSOURCE($x'$)
14:    $\mathcal{V} \leftarrow \mathcal{V} \cup \{x'\}$
15: **end for**
16: **return** $\mathcal{V}$

---

and (2) TypeScript type inference (Hellendoorn et al., 2018). ContraCode improves accuracy on both tasks. As a baseline self-supervised approach, we pre-train a RoBERTa model with the masked language modeling (MLM) loss on our augmented dataset, then fine-tune it on each downstream task. Contrastive pre-training with our compiler-based augmentations outperforms baseline supervised learning methods as well as MLM self-supervision. To probe the semantic content of representations learned with MLM, ContraCode, and a hybrid model combining both objectives, we evaluate zero-shot performance of code clone detection (Kamiya et al., 2002), a binary classification task that reveals that contrastive and hybrid representations are highly predictive of program functionality in-the-wild. Further, we find it is better to augment the large set of unlabeled programs during pre-training rather than augmenting smaller supervised datasets. As ContraCode makes no modifications to model architecture, we find that contrastive pre-training can be applied to diverse baselines while improving accuracy across the board.

We pre-train over a large corpus of methods extracted from popular GitHub repositories. The CodeSearchNet dataset collected by Husain et al. (2019) contains 1,843,099 JavaScript programs. Only 81,487 methods have both a documentation string and a method name. The asymmetry between labeled and unlabeled programs stems from JavaScript coding practices where anonymous functions are widespread. The pre-training dataset described in Section 3.1 is the result of augmenting CodeSearchNet's 1.8m programs.

## 4.1 IMPACT OF CONTRACODE PRE-TRAINING ON TYPE INFERENCE

JavaScript is a dynamically typed language, where variable types are determined at runtime based on the values they represent. However, annotating code with types helps tools flag possible bugs before runtime by statically detecting incompatible types. These annotations also help programmers document and understand code. However, maintaining type annotations is tedious. Type inference tools automatically predict variable types from context.

To *learn* to infer types, we use the same annotated dataset of TypeScript programs from Deep-Typer (Hellendoorn et al., 2018), without GitHub repos that were made private or deleted since publication. The training set consists of 15,570 TypeScript files from 187 projects with 6,902,642 total tokens. Validation and test sets are from held-out repositories. For additional supervision during training, additional types are inferred by static analysis to augment user-defined types as targets. All type annotations are removed from the input to the model. We evaluate a 2-layer Bidirectional LSTM, as used by DeepTyper, and a 6-layer Transformer, modified from RoBERTa to have a comparable parameter count. A 2-layer MLP head predicts types from the model's embedding of each token. We perform early stopping based on validation set top-1 accuracy.

Benefiting from pre-training is challenging because it requires knowledge transfer across dialects. Our models are pre-trained on JavaScript, not TypeScript. TypeScript supports a superset of the JavaScript grammar, adding type annotations and syntactic sugar that must be learned during fine-tuning. Further, the pre-training dataset consists of methods, while the DeepTyper dataset includes

Table 2: Type inference accuracy on TypeScript programs in the Hellendoorn et al. (2018) dataset. ContraCode (BiLSTM) outperforms baseline top-1 accuracies by 2.28% to 13.16%. As ContraCode does not modify model architecture, contrastive pre-training can be combined with each baseline. Compared with TypeScript's built-in type inference, ContraCode improves top-1 accuracy by 8.9%.

| Baseline | Method | Acc@1 (all types) | Acc@5 (all types) |
|---|---|---|---|
| Static analysis | TypeScript CheckJS (Bierman et al., 2014) | 45.11% | — |
| | Name only (Hellendoorn et al., 2018) | 28.94% | 70.07% |
| Transformer | Transformer (supervised) | 45.66% | 80.08% |
| | with ContraCode pre-training | **46.86%** | **81.85%** |
| RoBERTa | Transformer (RoBERTa MLM pre-training) | 40.85% | 75.76% |
| | with ContraCode pre-training | **47.16%** | **81.44%** |
| DeepTyper (BiLSTM) | DeepTyper (supervised) | 51.73% | 82.71% |
| | with RoBERTa MLM pre-training (10ᴋ steps) | 50.24% | 82.85% |
| | with ContraCode pre-training | 52.65% | 84.60% |
| | with ContraCode pre-training (w/ subword reg. ft.) | **54.01%** | **85.55%** |

entire modules. Table 2 summarizes results. Contrastive pre-training outperforms all baseline learned methods, showing meaningful transfer. Our best-performing model (bottom row) achieves +8.3% higher top-1 accuracy than a supervised Transformer model trained from scratch, +13.2% higher than a pre-trained RoBERTa model and +2.3% higher than DeepTyper.

ContraCode can also be applied in a drop-in fashion to each of the baselines without modifying model architecture. Simply pre-training each baseline with our contrastive objective and data augmentations yields absolute accuracy improvements of +1.2%, +6.3%, +2.3% top-1 and +1.8%, +5.7%, +2.8% top-5 over the Transformer, RoBERTa, and DeepTyper, respectively. The RoBERTa baseline may perform poorly since its masked language modeling (MLM) objective focuses on token reconstruction that is overly sensitive to local syntactic structure. To combine the approaches, we minimized our loss in addition to MLM as a hybrid local-global objective during pre-training.

Learning outperforms static analysis by a large margin. Overall, ContraCode achieves +8.9% higher top-1 accuracy than the best static type inference system, the built-in TypeScript CheckJS system, showing the promise of learned code analysis. Surfacing multiple candidate types can be useful to users. While CheckJS only produces a single prediction which is often incorrect, one of the top-5 predictions of ContraCode is correct for 85.55% of labeled tokens.

### 4.2 IMPACT OF CONTRACODE PRE-TRAINING ON EXTREME CODE SUMMARIZATION

The extreme code summarization task asks a model to predict the name of a method given its body (Allamanis et al., 2016). Tokenized method names often contain a short summary of functionality, such as `reverseString(...)`. Summarization models could explain obfuscated or poorly documented code. We create a JavaScript summarization dataset using the 81,487 labeled methods in the CodeSearchNet dataset. The method name is masked in the declaration of the function and then predicted by a sequence-to-sequence model with an autoregressive decoder trained to maximize log likelihood of the ground-truth name, a form of abstractive summarization. All models overfit, so we use early stopping according to validation loss. As proposed by Allamanis et al. (2016), we evaluate model predictions by precision, recall and F1 scores over the set of method name tokens.

Table 3 shows code summarization results in four settings: (1) supervised training using baseline tree-structured architectures that analyze the AST (code2vec, code2seq), (2) pre-training on all 1.84M programs using masked language modeling followed by fine-tuning on the labeled programs (RoBERTa), (3) supervised training from scratch with a Transformer architecture and (4) contrastive pre-training with all 1.84M programs followed by fine-tuning with augmentations (ContraCode).

Contrastive pre-training with fine-tuning outperforms the prior code2seq model, a competitive supervised baseline, by 8.2% in test precision, 7.3% in recall, and 7.9% in F1 score. The tree-based

Table 3: Results for different settings of the **code summarization task**: supervised training with 81k functions, masked language model pre-training and contrastive pre-training with fine-tuning.

| Method | Precision | Recall | F1 |
|---|---|---|---|
| code2vec (Alon et al., 2019b) | 10.78% | 8.24% | 9.34% |
| code2seq (Alon et al., 2019a) | 12.17% | 7.65% | 9.39% |
| RoBERTa MLM (Liu et al., 2019) | 15.13% | 11.47% | 12.45% |
| Transformer (Vaswani et al., 2017) | 18.11% | 15.78% | 16.86% |
| Transformer + ContraCode + augmentation | 20.34% | 14.96% | **17.24%** |

code2seq architecture is a way to encode code-specific invariances into the model, while contrastive pre-training induces domain invariances through data augmentation; reduced inductive biases in the Transformer model architecture leads to better performance. ContraCode outperforms self-supervised pre-training with RoBERTa by $4.8\%$ F1. ContraCode also achieves higher performance than the Transformer learned from scratch with the same network architecture. While this improvement is relatively smaller, code summarization is a difficult task. Naming conventions aren't consistent between programmers, and the metric measures exact token matches.

### 4.3 PROBING REPRESENTATIONS OF FUNCTIONALITY: ZERO-SHOT CODE CLONE DETECTION

ContraCode learns to match variants of programs with similar functionality. While these transformations produce highly diverse token sequences (Section A.4), they are artificial and do not change the underlying algorithm. Human programmers can solve a problem with many data structures, algorithms and programming models. Are pre-trained representations consistent across programs written by different people? We benchmark on the *code clone detection task*, a binary classification task to distinguish pairs of programs solving the same problem from pairs solving different ones. This is useful for deduplicating and refactoring code, or checking approximate code correctness.

Table 4: **Code clone detection** results with cosine similarity probe. Contrastive and hybrid representations are predictive of functionality, with $+6.2\%$, $+10\%$ AUROC over textual similarity (edit distance).

| Representation | AUROC | AP |
|---|---|---|
| Edit distance heuristic | 69.55 | 73.75 |
| Transformer w/o pre-train | 74.28 | 76.40 |
| + MLM pre-train | 74.41 | 75.96 |
| + ContraCode pre-train | 75.76 | 78.16 |
| + ContraCode + MLM | **79.55** | **81.74** |

Datasets exist like BigCloneBench (Svajlenko et al., 2014), but to the best of our knowledge, there is no benchmark for the JavaScript programming language. We collected 274 in-the-wild JavaScript programs correctly solving 33 problems from the HackerRank interview preparation website. There are 2065 pairs solving the same problem and 70K pairs solving different problems, which we randomly subsample to 2065 to balance the classes. Since we probe zero-shot performance, there is no training set. Traditional code analysis methods for clone detection measure textual similarity. As a baseline heuristic classifier, we threshold the dissimilarity score (Eq. 2), a scaled edit distance between two normalized and tokenized programs (to exclude formatting changes). For continuous representations, we threshold cosine similarity $u^T v/\|u\|\|v\|$.

Table 4 shows results according to the area under the ROC curve (AUROC) and average precision (AP, area under precision-recall). Continuous representation improves clone detection over the heuristic. However, self-supervision through masked language modeling for nearly 100 epochs of pre-training does not help, indicating that MLM is a poor fit for representing functionality. Contrastive pre-training achieves $+6.21\%$ higher AUROC than the baseline. A hybrid objective combining both the contrastive loss and MLM has the best performance with $+10\%$ AUROC ($+5.14\%$ over MLM alone).

### 4.4 UNDERSTANDING THE IMPORTANCE OF DATA AUGMENTATION

We first analyze the effect of our proposed augmentations on supervised learning without a pre-training phase. We then study the importance of individual augmentations during pre-training.

**Supervised learning with data augmentation** As a baseline, we re-train models from scratch with compiler transforms during *supervised learning* rather than pre-training. Data augmentation

Table 6: Ablating compiler transformations used during contrastive pre-training. The DeepTyper BiLSTM is pre-trained with constrastive learning for 20ᴋ steps, then fine-tuned for type inference. Augmentations are only used during pre-training. Each transformation contributes to accuracy.

| Augmentations used during pre-training | Acc@1 | Acc@5 |
|---|---|---|
| All augmentations (Table 2) | **52.65%** | **84.60%** |
|    without identifier modification (-VR, -IM) | 51.94% | 84.43% |
|    without line subsampling (-LS) | 51.05% | 81.63% |
|    without code compression (-T,C,DCE,CF) | 50.69% | 81.95% |

artificially expands labeled training sets. For sequence-to-sequence summarization, we apply a variety of augmentations; these all preserve the method name label. For type inference, labels are aligned to input tokens, so they must be realigned after transformation. We apply all token-level transformations that track label locations.

Table 5 shows results. Compiler-based data augmentations degrade supervised models, perhaps by creating a training distribution not reflective of evaluation programs. However, as shown in 4.1 – 4.3, augmenting during ContraCode pre-training yields a more robust model. Our contrastive learning framework also allows learning over large numbers of unlabeled programs that supervised learning alone cannot leverage. The ablation indicates that augmentations do not suffice, and contrastive learning is important.

**Ablating data pre-training augmentations** Some data augmentations may be more valuable than others for learn-

Table 5: Compiler data augmentations degrade performance when training supervised models *from scratch*.

| Code summarization | F1 |
|---|---|
| Transformer (Table 3) | **16.86** |
|   w/ LS,SW,VR,DCI aug. | 15.65 |

| Type Inference | Acc@1 |
|---|---|
| Transformer (Table 2) | **45.66** |
|   w/ SW reg. | 43.96 |
|   w/ LS,SW aug. | 44.14 |
| DeepTyper (Table 2) | **51.73** |
|   w/ SW reg. | 49.93 |
|   w/ LS,SW aug. | 50.93 |
|   w/ stronger LS,SW aug. | 50.33 |

ing a representation via instance discrimination. Empirically, pre-training converges faster with a smaller set of augmentations at the same batch size since the positives are syntactically more similar, but this hurts downstream performance. Table 6 shows that type inference accuracy degrades when different groups of augmentations are removed. Semantics-preserving code compression passes that require code analysis are the most important, improving top-1 accuracy by 1.95% when included. Line subsampling serves as a regularizer, but changes program semantics. LS is relatively less important, but does help accuracy. Identifier modification passes preserve semantics, but remove potentially useful naming information. Removing these hurts accuracy the least.

**Additional results** We perform additional ablations in Section A.1 by transferring different parts of the network to downstream tasks, computing the contrastive objective with representations taken from different encoder layers, varying architecture, and tuning the pre-training procedure. These experiments suggest that as many parameters as possible should be transferred to the downstream task. Details of the pre-training strategy are also important. Computing the contrastive objective using a "global" representation $q$ summarizing the whole input sequence $x^q$ outperforms more a "local" representation based on aggregating token representations. Further, a large batch size is helpful to stabilize pre-training. Section A.2 includes qualitative results.

## 5 CONCLUSIONS

Large-scale unannotated repositories of code like GitHub are a powerful resource for learning machine-aided programming tools. However, most current approaches to code representation learning do not leverage unannotated data. We propose ContraCode, a contrastive self-supervised algorithm that learns representations that are invariant to code transformations. Our method optimizes for this invariance via novel compiler-based data augmentations for code. ContraCode significantly improves the accuracy of extreme code summarization baselines (+2.3% to +13.2%), TypeScript type inference models (up to +7.9% F1) and code clone detection (+5 to +10% AUROC). ContraCode outperforms self-supervised RoBERTa pre-training. Moreover, contrastive pre-training outperforms supervised training with our augmentations. As ContraCode makes no modifications to model architecture and simply adds a training phase, it consistently improves accuracies when applied to diverse baselines.

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

# A APPENDIX

## A.1 ADDITIONAL RESULTS AND ABLATIONS

**Code clone detection ROC and PR curves** Figure 5 plots true postive rate vs false positive rate and precision vs recall for different zero-shot classifiers on the code clone detection downstream tasks. These classifiers threshold a similarity score given by token-level edit distance for the heuristic approach or cosine similarity for the neural network representations. The hybrid self-supervised model combining ContraCode's contrastive objective and masked language modeling achieves better tradeoffs than the other approaches.

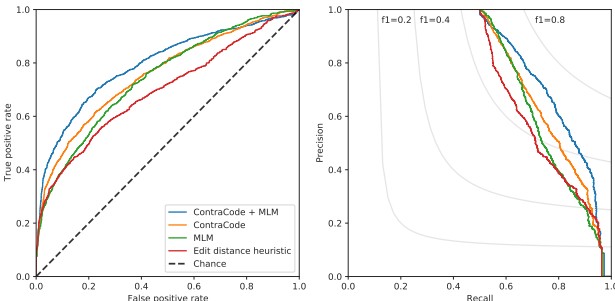

Figure 5: Receiver Operating Characteristic (ROC, left) and Precision-Recall (PR, right) curves for zero-shot classifiers on the code clone detection task. Equal F1 score curves are shown on right.

**Which part of the model should be transferred?** SimCLR (Chen et al., 2020a) proposed using a small MLP head to reduce the dimensionality of the representation used in the InfoNCE loss during pre-training, and did not transfer the MLP to the downstream image-classification task. In contrast, we find it beneficial to transfer part of the contrastive MLP head to type inference, showing a $2\%$ improvement in top-5 accuracy over transferring the encoder only (Table 7). We believe the improvement stems from fine-tuning both the encoder and MLP which allows feature adaptation, while SimCLR trained a linear model on top of frozen features. We only transferred the MLP when contrasting the mean of token embeddings during pre-training, not the terminal hidden states, as the dimensionality of the MLP head differs. These representations are compared next.

Table 7: If local representations are learned, transferring part of the Contrastive MLP head improves type inference. The encoder is a 2-layer BiLSTM (d=512), with a 2-layer MLP head for both pre-training purposes and type inference. The mean hidden state representation is optimized for 10K iterations for the purposes of this ablation.

| Transferred from pre-training | Acc@1 | Acc@5 |
|---|---|---|
| Transfer BiLSTM | **49.32%** | 80.03% |
| Transfer BiLSTM, 1 layer of MLP | 49.15% | **82.58%** |

**Should we pre-train global or local representations?** We compare pre-training DeepTyper with two variants of ContraCode. We either use the mean of token hidden states across the program (averaging local features), or the terminal hidden states as input to the MLP used to extract the contrastive representation $q = f_q(x)$ (global features). Token-level features might capture more syntactic details, but averaging pooling ignores order. Table 8 shows the accuracy of a BiLSTM pre-trained with each strategy. Using the global features for pre-training yields significantly improved performance, +2.38% acc@1 after 10K iterations of pre-training (not converged for the purposes of ablation). The global pre-training strategy achieves the best results in Table 2.

**Do pre-trained encoders help more with shallow decoders?** For the sequence-to-sequence code summarization task, ContraCode only pre-trains the encoder of the Transformer. In Table 9, we ablate the depth of the decoder to understand how much shallow decoders benefit from contrastive pre-training of the encoder. Similar experiments were performed in a vision context by Erhan et al.

Table 8: Contrasting global, sequence-level representations outperforms contrasting local representations. We compare using the terminal (global) hidden states of the DeepTyper BiLSTM and the mean pooled token-level (local) hidden states.

| Representation | Optimization | Acc@1 | Acc@5 |
|---|---|---|---|
| Global | InfoNCE with terminal hidden state, 20K steps (Table 2) | **52.65%** | **84.60%** |
| | InfoNCE with terminal hidden state, 10K steps | 51.70% | 83.03% |
| Local | InfoNCE with mean token rep., 10K steps | 49.32% | 80.03% |

Table 9: Training time and decoder depth ablation on the method name prediction task. Longer pre-training significantly improves downstream performance when a shallow, 1 layer decoder is used.

| Decoder | Pre-training (1.8M programs) | Supervision (81k programs) | Precision | Recall | F1 |
|---|---|---|---|---|---|
| Transformer, 1 layer | MoCo, 10k steps | Original set | 11.91% | 5.96% | 7.49% |
| Transformer, 1 layer | MoCo, 45k steps | Original set | **17.71%** | **12.57%** | **13.79%** |
| Transformer, 4 layers | MoCo, 45k steps | Original set | **18.21%** | **13.21%** | **14.56%** |

(2010), where different numbers of layers of a classifier are pretrained. After 45k pre-training steps, the 4-layer decoder achieves 0.50% higher precision, 0.64% higher recall and 0.77% higher F1 score than the 1-layer model, so additional decoder depth is helpful for the downstream task. The 1-layer decoder model also benefits significantly from longer pre-training, with a 6.3% increase in F1 from 10k to 45k iterations. This large of an improvement indicates that ContraCode could be more helpful for pre-training when the number of randomly initialized parameters at the start of fine-tuning is small. For larger decoders, more parameters must be optimized during-finetuning, and the value of pre-training is diminished.

**Contrastive representation learning strategies** In Figure 6, we compare two strategies of refreshing the MoCo queue of key embeddings (the dictionary of negative program representations assumed to be non-equivalent to the batch of positives). In the first strategy, we add 8 items out of the batch to the queue ($1\times$), while in the second we add 96 items ($12\times$). In addition, we use a larger queue (65k versus 125k keys) and a slightly larger batch size (64 versus 96). We observe that for the baseline queue fill rate, the accuracy decreases for the first 8125 iterations as the queue fills. This decrease in accuracy is expected as the task becomes more difficult due to the increasing number of negatives during queue warmup. However, it is surprising that accuracy grows so slowly once the queue is filled. We suspect this is because the key encoder changes significantly over thousands of iterations: with a momentum term $m = 0.999$, the original key encoder parameters are decayed by a factor of $2.9 \times 10^{-4}$ by the moving average. If the queue is rapidly refreshed, queue embeddings are predicted by recent key encoders, not old parameters. This also indicates that a large diversity of negative, non-equivalent programs are helpful for rapid convergence of ContraCode pre-training.

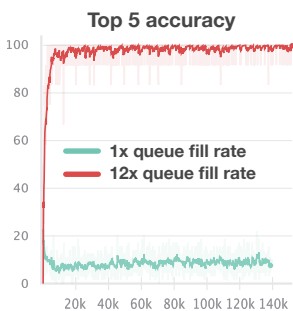

Figure 6: Pre-training quickly converges if negative programs in the queue are frequently changed.

## A.2 QUALITATIVE RESULTS

**t-SNE visualization of representations** We qualitatively inspect the structure of the learned representation space by visualizing self-supervised representations of variants of 28 programs using t-SNE (Maaten & Hinton, 2008) in Figure 7. Representations of transformed variants of the same program are plotted with the same color. ContraCode (BiLSTM) clusters variants closely together. Indeed, contrastive learning learns representations that are invariant to a wide class of automated compiler-based transformations. In comparison, the representations learned by masked language modeling (RoBERTa) show more overlap between different programs, and variants do not cleanly cluster.

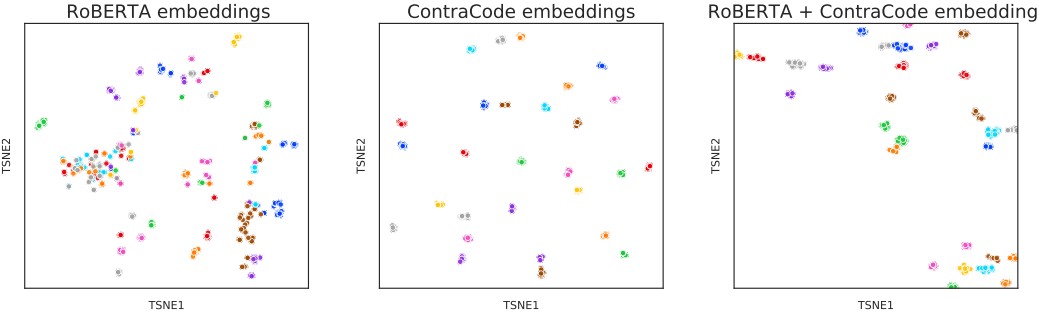

Figure 7: t-SNE (Maaten & Hinton, 2008) plot of program representations learned with masked language modeling (RoBERTa), contrastive learning (ContraCode), and a hybrid loss (RoBERTa + ContraCode). Transformed variants of the same program share the same color, though colors may be similar across different programs.

With a hybrid loss combining masked language modeling and contrastive learning, representations of variants of the same program once again cluster.

**Code summaries** Figure 8 shows a qualitative example of predictions for the code summarization task. The JavaScript method is not seen during training. A Transformer pretrained with ContraCode predicts the correct method name as the most likely decoding through beam search. The next four predictions are reasonable, capturing that the method processes an image. The 2nd and 3rd most likely decodings, `getImageItem` and `createImage`, use `get` and `create` as synonyms for `load`, though the final two unlikely decodings include terms not mentioned in the method body.

```javascript
function x(url, callback, error) {
  var img = new Image();
  img.src = url;
  if(img.complete){
    return callback(img);
  }
  img.onload = function(){
    img.onload = null;
    callback(img);
  };
  img.onerror = function(e){
    img.onerror = null;
    error(e);
  };
}
```

*Ground truth*: `loadImage`
*Prediction:* `loadImage`

Other predictions:

1. `getImageItem`
2. `createImage`
3. `loadImageForBreakpoint`
4. `getImageSrcCSS`

Figure 8: A JavaScript program from the CodeSearchNet dataset not seen during training and the predicted method names from a Transformer pre-trained with ContraCode. ContraCode predicts the correct method name as its most likely decoding.

**Type inferences** We can also visualize outputs of the type inference model. Figure 9 shows two TypeScript programs from the held-out test set. User-provided type annotations are removed from the programs, and the model is provided with a tokenized form without access to dependencies. We visualize predictions from a variant of DeepTyper pretrained with ContraCode, the best-performing model in Table 8. In the first program, our model consistently predicts the correct return and parameter type. While a tool based on static analysis could infer the `void` return types, the type of the `message` argument is ambiguous without access to the imported `write` method signature. Still, the model correctly predicts with high confidence that the variable `message` is a string. In the second program, ContraCode correctly predicts 4 of 8 types including the `ViewContainerRef` and `ChangeDetectorRef` types, each imported from the AngularJS library. As this sample is held-out from the training set, these predictions show generalization from other repositories using AngularJS.

```
import {
  write,
  categories,
  messageType
} from "s";
export const animationsTraceCategory = "s";
export const rendererTraceCategory = "s";
export const viewUtilCategory = "s";
export const routerTraceCategory = "s";
export const routeReuseStrategyTraceCategory = "s";
export const listViewTraceCategory = "s";
export function animationsLog ( message: string 100.0% ): void 99.9% {
  write(message, animationsTraceCategory);
}
export function rendererLog (msg): void 53.7% {
  write(msg, rendererTraceCategory);
}
export function rendererError ( message: string 99.5% ): void 99.7% {
  write(message, rendererTraceCategory, messageType.error);
}
export function viewUtilLog (msg): void 100.0% {
  write(msg, viewUtilCategory);
}
export function routerLog ( message: string 99.9% ): void 100.0% {
  write(message, routerTraceCategory);
}
export function routeReuseStrategyLog ( message: string 99.8% ): void 99.98% {
  write(message, routeReuseStrategyTraceCategory);
}
export function styleError ( message: string 99.97% ): void 100.0% {
  write(message, categories.Style, messageType.error);
}
export function listViewLog ( message: string 100.0% ): void 100.0% {
  write(message, listViewTraceCategory);
}
export function listViewError ( message: string 99.93% ): void 100.0% ...
```

```
import {
  ComponentRef,
  ComponentFactory,
  ViewContainerRef,
  Component,
  Type,
  ComponentFactoryResolver,
  ChangeDetectorRef
} from "s";
import {
  write
} from "s";
export const CATEGORY = "s";

function log( message: string 56.95 ) {
  write(message, CATEGORY);
}
@ Component({
  selector: "s",
  template: `template`
}) export class DetachedLoader {
  constructor(private resolver: ViewContainerRef 63.85% (GT: ComponentFactoryResolver) ,
              private changeDetector: ChangeDetectorRef 100.0% ,
              private containerRef: ViewContainerRef 100.0% ) {}
  private loadInLocation (
      componentType<any>: TemplateRef 99.6% (GT: Type)) <ComponentRef<any>>: Promise 100.0% {
    const factory = this.resolver.resolveComponentFactory(componentType);
    const componentRef = this.containerRef.createComponent(
      factory, this.containerRef.length, this.containerRef.parentInjector);
    log("s");
    return Promise.resolve(componentRef);
  }
  public detectChanges() {
    this.changeDetector.markForCheck();
  }
  public loadComponent (
      componentType<any>: TemplateRef 99.9% (GT: Type)) <ComponentRef<any>>: Promise 100.0% {
    log("s");
    return this.loadInLocation(componentType);
  } ...
```

Figure 9: Our model, a variant of DeepTyper pretrained with ContraCode, generates type annotations for two programs in the held-out set. The model consistently predicts the correct return type of functions, and even predicts project-specific types imported at the top of the file. The model corresponds to the top row of Table 8, though is not our best performing model.

### A.3 PROGRAM TRANSFORMATION DETAILS

We use the Babel compiler infrastructure (McKenzie et al., 2020) and the `terser` JavaScript library for AST-based program transformations. We perform variable renaming and dead code insertion (variable declaration insertion) using custom Babel transforms, subword regularization with `sentencepiece` Python tokenization library, line subsampling using JavaScript string manipulation primitives and other transformations with `terser`. Terser has two high-level transformation modes, mangling and compression, each with finer grained controls such as formatting, comment and log removal, and dead code elimination. We show an example merge sort with example equivalent variants in Figure 11.

**Reformatting, beautification, compression (R, B, C):** Personal coding conventions do not affect the semantics of code; auto-formatting normalizes according to a style convention.

**Dead-code elimination (DCE):** In this pass, all unused code with no side effects are removed. Various statements can be inlined or removed as stale or unneeded functionality.

**Type upconversion (T):** In JavaScript, some types are polymorphic & can be converted between each other. As an example, booleans can be represented as `true` or as `1`.

**Constant folding (CF):** During constant folding, all expressions that can be pre-computed at compilation time can be inlined. For example, the expression `(2 + 3) * 4` is replaced with `20`.

**Variable renaming, identifier mangling (VR, IM):** Arguments can be renamed with random word sequences and identifiers can be replaced with short tokens to make the model robust to naming choices. Program behavior is preserved despite obfuscation.

**Dead-code insertion (DCI):** Commonly used no-ops such as comments and logging are inserted.

**Subword regularization (SW):** From Kudo (2018), text is tokenized in several different ways, with a single word (`_function`) or subtokens (`_func tion`).

**Line subsampling (LS):** We randomly sample ($p = 0.9$) lines from a method body. While not semantics-preserving, line subsampling serves as a regularizer.

While compilers are generally deterministic, we require a variety of alternatives to each program for contrastive representation learning. Algorithm 1 samples $N$ augmented variants of a source program $x$ using a set of deterministic compiler transformations $\tau_i$. Stochasticity is introduced by randomly toggling each transformation according to Bernoulli samples with probabilities $p_i$. When adding a program to the set of variants $\mathcal{V}$, uniqueness is determined by string comparison.

### A.4 HOW SIMILAR ARE TRANSFORMED PROGRAMS?

To understand the diversity created by program transformations, we compute the Levenshtein minimum edit distance between positive pairs in the precomputed pre-training dataset, *i.e.* transformed variants of the same source method. For comparison, we also compute the edit distance between negative pairs, *i.e.* transformed variants of different programs. The edit distance $D(x_q, x_k)$ computes the minimum number of token insertions, deletions or substitutions needed to transform the tokenized query progrm $x_q$ into the key program $x_k$. To normalize by sequence length $|\cdot|$, let

$$\text{dissimilarity}_D(x_q, x_k) = \frac{D(x_q, x_k)}{\max(|x_q|, |x_k|)} \tag{2}$$

Dissimilarity ranges from 0% for programs with the same token sequence such as positives before applying our transformations, to 100% for programs without any shared tokens. Note that whitespace transformations do not affect the metric because the tokenizer collapses repeated whitespace. For the positives, we estimate dissimilarity by sampling one pair per source program in the CodeSearchNet dataset (1.6M source programs with at least one pair). We sample the same number of negative pairs.

Figure 10 shows a histogram of token dissimilarity. Positive pairs have $65\%$ mean dissimilarity, while negatives have $86\%$ mean dissimilarity. Negatives are more dissimilar on average as source sequences could have different lengths, idioms and functionality. Still, the transformations generated quite different positive sequences, with less than half of their tokens shared. The 25th, median and 75th percentile dissimilarity is 59%, 66% and 73% for positives, and 82%, 87% and 90% for negatives.

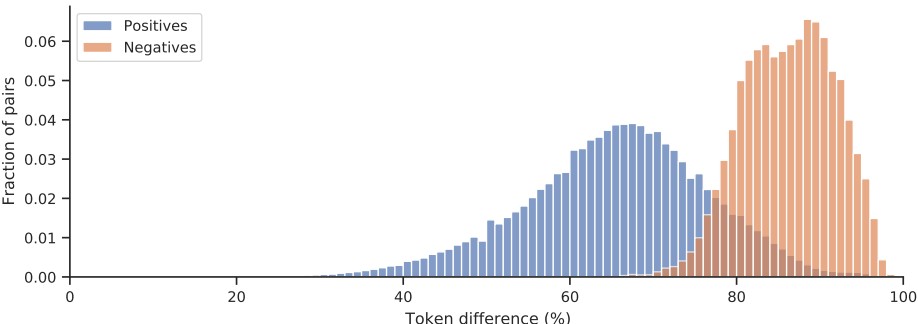

Figure 10: Histogram of pairwise token dissimilarity for contrastive positives (transformed variants of the same method) and negatives (transformed variants of different methods). Code transformations produce positives with dissimilar token sequences.

## A.5 EXPERIMENTAL SETUP

**Architectures** The Transformer has 6 encoder layers (23M parameters) in all experiments, and 4 decoder layers for method name prediction in Table 3. We leverage the default positional embedding function (sin, cos) as used in the original Transformer architecture. The network originally proposed in DeepTyper (Hellendoorn et al., 2018) had 11M parameters with a 300 dimensional hidden state. We increase the hidden state size to 512 to increase model capacity, so our BiLSTM for type prediction has 17.5M parameters. During fine-tuning, across all experiments, we optimize parameters using Adam with linear learning rate warmup and decay. For the Transformer, the learning rate is linearly increased for 5,000 from 0 to a maximum of $10^{-4}$. For the bidirectional LSTM, the learning rate is increased for between 2,500 and 10,000 steps to a maximum of $10^{-3}$. Type inference hyperparameters are selected by validation top-1 accuracy.

**ContraCode pretraining** The InfoNCE objective (1) is minimized with temperature $t = 0.07$ following He et al. (2019). Also following He et al. (2019), the key encoder's parameters are computed with the momentum update equation $\theta_k \leftarrow m\theta_k + (1 - m)\theta_q$, equivalent to an EMA of the query encoder parameters $\theta_q$. To pretrain a Transformer using the ContraCode objective, we first embed each token in the program using the Transformer. However, the InfoNCE objective is defined in terms of a single embedding for the full program. The ContraCode Transformer is pretrained with a batch size of 96. Our model averages the 512-dimensional token embeddings across the sequence, then applies a two-layer MLP with 512 hidden units and a ReLU activation to extract a 128-dimensional program embedding for the loss.

The DeepTyper bidirectional LSTM architecture offers two choices for extracting a global program representation. We aggregate a 1024-dimensional global representation of the program by concatenating its four terminal hidden states (from two sequence processing directions and two stacked LSTM layers), then apply the same MLP architecture as before to extract a 128-dimensional program representation. Alternatively, we can average the hidden state concatenated from each direction across the tokens in the sequence before applying the MLP head. We refer to the hidden-state configuration as a global representation and the sequence averaging configuration as a local representation in Table 8. We pre-train the BiLSTM with large batch size of 512 and apply weight decay.

**Type prediction** Following DeepTyper (Hellendoorn et al., 2018), our regenerated dataset for type prediction has 187 training projects with 15,570 TypeScript files, totaling 6,902,642 tokens. We tune hyperparameters on a validation set of 23 distinct projects with 1,803 files and 490,335 tokens, and evaluate on a held-out test set of 24 projects with 2,206 files and 958,821. The training set is smaller than originally used in DeepTyper as several projects were made private or deleted from GitHub before May 2020 when we downloaded the data, but we used the same commit hashes for available projects so our splits are a subset of the original. We have released the data with our open-source code to facilitate further work on a stable benchmark as more repositories are deleted over time. We perform early stopping to select the number of training epochs. We train each model for 100 epochs and select the checkpoint with the minimum accuracy@1 metric (all types, including

```javascript
// Split the array into halves and merge them recursively
function mergeSort (arr) {
  if (arr.length === 1) {
    // return once we hit an array with a single item
    return arr
  }
  const middle = Math.floor(arr.length / 2)
  // get the middle item of the array rounded down
  const left = arr.slice(0, middle)
  // items on the left side
  const right = arr.slice(middle)
  // items on the right side
  return merge(
    mergeSort(left),
    mergeSort(right)
  )
}
```

**Original merge sort program**

```javascript
function mergeSort(e) {
  if (e.length === 1) {
    return e;
  }
  const t = Math.floor(e.length / 2);
  const l = e.slice(0, t);
  const n = e.slice(t);
  return merge(mergeSort(l), mergeSort(n));
}
```

**After variable renaming, comment removal, reformatting** (mangling)

```javascript
function mergeSort(e) {
  if (1 === e.length) return e;
  const t = Math.floor(e.length / 2), r = e.slice(0, t), n = e.slice(t);
  return merge(mergeSort(r), mergeSort(n));
}
```

**After combining variable declarations, inlining conditional** (mangling and compression)

Figure 11: Given a JavaScript code snippet implementing the merge sort algorithm, we apply semantics-preserving transformations to produce functionally-equivalent yet textually distinct code sequences. Compression passes eliminates unnecessary characters such as redundant variable declarations and brackets, while mangling passes can change variable names.

any) on the validation set. Except for the model learned from scratch, the Transformer architectures are pre-trained for 240K steps. Models with the DeepTyper architecture converge faster on the pre-training tasks and are pre-trained for 20K iterations (unless otherwise noted).

**Extreme code summarization by method name prediction** We train method prediction models using the labeled subset of CodeSearchNet. Neither method names nor docstrings are provided as input to the model: the docstring is deleted, and the method name is replaced with the token 'x'. Thus, the task is to predict the method name using the method body and comments alone. To decode method names from all models except the code2vec and code2seq baselines which implement their own decoding procedures, we use a beam search with a beam of size 5 and a maximum target sequence length of 20 subword tokens. We detail the cumulative distribution of program lengths in Figure 12. The ContraCode summarization Transformer only needed to be pre-trained for 20K iterations, with substantially faster convergence than RoBERTa (240K iterations). During fine-tuning, we apply the LS,SW,VR,DCI augmentations to ContraCode.

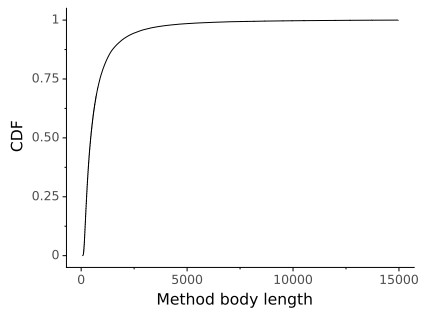
(a) Character length per code sample

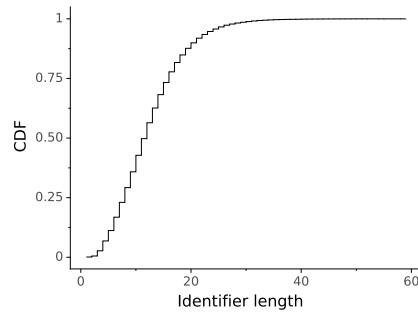
(b) Character length per method name

Figure 12: CodeSearchNet code summarization dataset statistics: (a) The majority of code sequences are under 2000 characters, but there is long tail of programs that span up to 15000 characters long, (b) JavaScript method names are relatively short compared to languages like C$^\sharp$ and Java.

## A.6 BASELINES

Baselines for code summarization and type prediction trained their models on an inconsistent set of programming languages and datasets. In order to normalize the effect of datasets, we selected several diverse state-of-the-art baselines and reimplemented them on the JavaScript dataset.

**AST-based models**   The authors of code2vec (Alon et al., 2019b) and code2seq (Alon et al., 2019a), AST-based code understanding models, made both data and code available, but train their model on the Java programming language. In order to extend the results in their paper to JavaScript for comparison with our approach, we generated an AST path dataset for the CodeSearchNet dataset. The sensitivity of path-mining embeddings to different datasets is documented in prior work, so published F1 scores are not directly comparable; F1 scores for code2vec (Alon et al., 2019b) vary between 19 (Alon et al., 2019a) and 43 (Alon et al., 2019b) depending on the dataset used. Therefore, we use the same dataset generation code as the authors for fair comparison. We first parse the source functions using the Babel compiler infrastructure. Using the original code on these ASTs, up to 300 token-to-token (leaf-to-leaf) paths are extracted from each function's AST as a precomputed dataset. Then, we generate a token and AST node vocabulary using the same author-provided code, and train the models for 20 epochs, using early stopping for code2seq. We observed that code2vec overfits after 20 epochs, and longer training was not beneficial.

**DeepTyper (Hellendoorn et al., 2018)**   DeepTyper uses a two layer GRU with a projection over possible classes, with an embedding size of 300 and hidden dimension of 650. However, we found improved performance by replacing the GRU with a bidirectional LSTM (BiLSTM). We normalize the LSTM parameter count to match our model, and therefore use a hidden dimension size of 512. We also use subword tokenization rather than space delimited tokens according to Kudo (2018), as subword tokenization is a key part of state-of-the-art models for NLP (Sennrich et al., 2015).

**RoBERTa**   We pre-trained an encoder using RoBERTa's masked language modeling loss on our augmented version of CodeSearchNet, the same data used to pretrain ContraCode. This model is then fine-tuned on downstream datasets. Unlike the original BERT paper which cuBERT (Kanade et al., 2020) is based on, hyperparameters from RoBERTa have been found to produce better results during pre-training. RoBERTa pre-trains using a masked language modeling (MLM) objective, where 15% of tokens in a sentence are masked or replaced and are reconstructed by the model. We did not use the BERT Next Sentence Prediction (NSP) loss which RoBERTa finds to be unnecessary. We normalize baseline parameter count by reducing the number of Transformer layers from 24 to 6 for a total of 23M parameters.

