# OpenReview forum: "Contrastive Code Representation Learning"
_ICLR.cc/2021/Conference — Reject_

### Official Review · AnonReviewer1 · 2020-10-27
**Interesting paper with some debatable claims**

**Rating:** 6
**Confidence:** 4

**Review:**

## Summary

The paper proposes Contrastive Code Representation Learning (ContraCode), a
self-supervised algorithm to learn task-agnostic semantic representations of
programs via contrastive learning. The guiding principle for contrastive
learning is that programs with the same functionality should have the same
representation. The authors develop an automated source-to-source compiler to
generate different (most of the time equivalent) variants of the same program.
The neural network, which is basically identical to the Momentum Contrast
architecture, is trained using the programs generated by this compiler.

## Pros

- A very nice and intuitive application of Momentum Contrast to code
  representation learning.
- The use of source-to-source compilers for contrastive learning.
- The results are consistently better than previous state-of-the-art on two
  different downstream tasks.
- The authors have done extensive ablation studies which provide further
  insight.

## Concerns

- I believe that the most emphasized result -- 40% higher top-5 accuracy than
  the current state-of-the-art static type analyzer for TypeScript -- is
  misleading. As far as I know, TypeScript's built-in type inference returns
  only a single suggestion. This is reinforced by the fact that Acc@1 and Acc@5
  is exactly the same for CheckJS in Table 2. So comparing Acc@5 between the two
  algorithms is not fair and it is pointless.
- In Section 3.2, the extent to which He et al. (Momentum Contrast) is followed
  is not clear enough. Part of the method that's described in "Pre-training
  objective" strictly follows/summarizes Momentum Contrast but that's not
  apparent from the paper.
- In 4.1., I find "cross-language knowledge transfer" a bit of an overstatement,
  as TypeScript is a superset of JavaScript.
- In 4.2., the difference in F1 score between Transformer and
  Transformer+ContraCode+augmentation is very small. I would like the authors to
  discuss this.
- In 4.2., I find the statement "showing that code token reconstruction is not
  an effective pre-training strategy" too general.

## Questions

I have questions about the transformations done with the source-to-source
compiler, which could also be made clearer in the paper.

Figure 3 shows the uniqe transformed program variants after applying 20
sequences of random transformations.
How many transformations were done in a sequence?

In 3.2, the authors write that each program is transformed twice (similarly to
Momentum Contrast). However, in Section 4, Pre-training dataset, they write that
the augmented dataset is pre-computed by sampling up to 20 unique transformed
variants per program.

Does this mean that the two transformed programs are sampled from these 20?
How many transformations were done in a sequence? Is that the same as for Figure 3?

## Reasons for Ranking

My main concern is the +40% Acc@5 claim compared to CheckJS, which I find
misleading. However, I think that this is an interesting paper and a valuable
contribution. If my concerns are addressed I'm willing to improve my score.

---

> ### Author Response · Authors · 2020-11-14
> **Thank you for the comments R1**
>
> Thanks for your helpful feedback! We have made changes accordingly.
>
> **Comparing Acc@5 between TypeScript CheckJS and your approach is not fair**
> **A:** We agree that this is a valid concern and have updated the draft with top-1 accuracy instead. We would like to point out that ContraCode is strong when evaluated with top-1 accuracy, with a 9% accuracy improvement over CheckJS and 2.3% over the best ML-based type inference baseline in Table 2. We originally presented top-5 as we feel there is value in surfacing multiple suggestions to users (like IDE autocompletion).
>
> **The extent to which He et al. is followed in Sec. 3.2 pre-training objective is not clear enough**
> **A:** We updated Sec. 3.2 to clarify that the InfoNCE loss is unmodified from He et al. Our approach is novel in its applications and modifications to support the language domain (such as textual data augmentation and encoder network architecture). For example,
>     - The pooling performed by He et al. substantially degrades accuracy with the LSTM architecture (Table 7, -2.3%-4.5%).
>     - The encoder representation for both the LSTM and Transformer backbones is 40x larger than in MoCo due to long sequence length, which created memory consumption challenges.
>     - An ablation study revealed MoCo’s queue contained stale representations that impacted convergence; we needed to increase queue fill rate for pre-training to converge (Figure 8).
>
> **In Sec. 4.1, cross-language knowledge transfer a bit of an overstatement**
> **A:** We acknowledge that we have not solved the cross-language transfer problem in general (e.g. very different syntaxes like JavaScript to Python), so we removed the “cross-lingual” label. However, TypeScript programs in the test set often look quite different from JavaScript. The TypeScript task also measures how our method transfers from per-method to cross-method representations as DeepTyper annotations are at a module level.
>
> **Small difference in F1 score in Sec. 4.2 for code summarization**
> **A:** Code summarization is a hard task. There is no standardized way to write a method name, so the same method can have multiple possible correct names. The summarization dataset is also relatively large (auto-generated) so pre-training shows smaller gains when compared with tasks like type inference, where our method shows a larger improvement. We added more discussion to Sec 4.2 regarding F1 score.
>
> **Sec. 4.2 statement “showing that code token reconstruction is not an effective pre-training strategy for code summarization” is too general**
> **A:** We removed it and now only report the quantitative improvement of ContraCode over RoBERTa. We’re working on more visualizations to inform our commentary on the differences between pre-training strategies.
>
> **Augmentation clarifications**
> **A:** To make the procedure clearer, we moved Algorithm 1 into the main body (from Appendix A.3) and edited the text. The whole pre-training dataset is augmented ahead of time according to Alg. 1, with up to 20 augmented variants for each program (setting $N=20$ in Alg. 1). Alg. 1 applies at most 11 transformations; each of the 11 is applied with some probability $p_i$ ($2^{11}$ possible sequences of transformations). Each variant might have a different number of transformations applied in sequence, from 0 to 11. The resulting dataset contains a list of alternatives for each program (statistics shown in Figure 3). During contrastive pre-training, we sample two variants from each of these lists.
>
> Hopefully this helps. We’re happy to answer more questions if not; please respond if there are any clarifications or further experiments we can provide.

---

> > ### Comment · AnonReviewer1 · 2020-11-21
> > **Thank you for your detailed answers and changes**
> >
> > Thank you for your answers and modifications.
> > I think they have made the paper much clearer.
> >
> > **I increased my score to 6.**
> >
> > I have one more concern: you write that "Pre-training with ContraCode consistently improves the F1 score of code summarization baselines by up to 8 percent". However, looking at Table 3, I think that much of the improvement is from the Transformer architecture, and less is from ContraCode.

---

> > > ### Author Response · Authors · 2020-11-25
> > > **Thank you!**
> > >
> > > We really appreciate your suggestions which helped strengthen our paper further, and thank you for raising your score. We revised the text based on your latest feedback. Since our earlier response to your review, we added a new code clone detection benchmark (originally requested by R2), which should add insight into the value of ContraCode beyond the Transformer architecture. The code clone results show larger gains with self-supervised learning with ContraCode, outperforming the Transformer that is not pre-trained: +5.27% in AUROC and +5.34% AP over Transformer.

---

### Official Review · AnonReviewer2 · 2020-10-27
**Contrastive Learning on Source Code Representation**

**Rating:** 6
**Confidence:** 3

**Review:**

This work proposes to combine contrastive learning with code representation. The different transformations of code snippets are inspried by static complier.

My mainly concern is about the novelty. I agree with the claim about programs with the same functionality should have the same underlying representation. However, it's unclear to me that why using it as contrastive loss is a better choice than MLM loss in code understanding downstream tasks, especially for type inference. Any theoretical or intuitive explaination is good.

It seems that the performance gain about the proposed method is overcliamed, especially for the 40\% top-5 accuracy gain of TypeScript which is a deterministic method. The actual gain compared to SOTA learning-based method is less than 3\%. Also, the experimental results are unconvincing to me, for example, pre-training with MLM loss (then finetune on the downstream task? The corresponding descriptions are not clear) get poor accuracy on type inference task. The authors intuitively explain it as that MLM loss is not  suitable for this kind of task. However, it performs better when combining with contrastive loss than only using contrastive pre-training.  Whether on earth MLM loss is good for this task?

Some necessary baseline methods are missing. For example, [1] and [2] for code summerazation task . And some important downstream tasks are also missing to demonstrate the ability of proposed method in code understanding, e.g., code clone detection (which I personally think that is more suitable for contrastive code representation).

[1] A Transformer-based Approach for Source Code Summarization, ACL 2020
[2] https://github.com/microsoft/CodeXGLUE/tree/main/Code-Text/code-to-text

---

> ### Author Response · Authors · 2020-11-18
> **Thanks for the feedback R2**
>
> We appreciate your questions. We added a new result (t-SNE visualizations) to the paper, and provide intuition below.
>
> **Main concern: intuition why contrastive loss or contrastive + MLM loss is better than MLM loss alone**
> **A:** Based on your questions, in Appendix A.2, Fig. 7, we updated the t-SNE visualization to qualitatively compare the representation learned with the MLM loss, contrastive loss and the hybrid loss. For each loss/subplot, 28 programs are transformed several times. Each transformed variant is embedded and shown in the same color.
>
> The t-SNE visualization reveals that the MLM loss (RoBERTa) learns representations that are highly sensitive to changes in the syntax as a result of code transformations. Transformed variants of the same program are not clustered. However, the contrastive loss tightly clusters programs with similar functionality; contrastive representations appear to be more robust to automated surface-level changes in the text of a program.
>
> MLM performs reconstruction, $\max_\theta \log p_\theta(x | \tilde{x})$: can the model predict the original program $x$ given a masked version $\tilde{x}$? The MLM loss weights all tokens equally as there’s no domain knowledge of what tokens are more important. For example, given program `print(“Hello ICLR”)` and sampled masks `[MASK](“Hello ICLR[MASK])`, MLM penalizes mispredictions of `print` and `”` equally. `print` is semantically meaningful, but `”` is an irrelevant detail. To minimize loss, the model needs to encode *all information* (lossless). This requires a lot of model capacity (20x more parameters than contrastive in [3]). It also has minor issues like a train-finetune domain shift (no masks during finetuning) and assumes conditional independence between masked tokens.
>
> Contrastive learning lets us specify domain knowledge of what bits are important in the sequence through our choice of transformations. It’s more like a lossy compressor [4]. The objective encourages the model to ignore irrelevant tokens that the compiler changes, such as switching `”` to `’`, so the model focuses on details that change functionality.
>
> *Hybrid loss (contrastive + MLM):* Functional embeddings are helpful, but functionality isn't the only information that matters for type inference. Even if inefficient, MLM can learn to encode some important details like variable names. That could explain why the hybrid loss performs well. In Fig. 7, a hybrid loss has the same tight clustering as a contrastive loss alone, but also learns some inter-cluster structure. In [5], contrastive learning combined with MLM yields a better score, and removing MLM from the method degrades the performance.
>
> **Overclaimed: should highlight smaller improvements over learned baselines rather than over TypeScript baseline.**
> **A:** We updated the abstract and intro: we now highlight the gain over learning approaches alongside the top-1 accuracy gain over TypeScript CheckJS. CheckJS is a relevant baseline for motivating learned code analysis over hand-designed analysis, so we think it adds context for readers not familiar with learned type inference. Let us know if you think it’s still overclaimed, we’re open to editing.
>
> **Code summarization baselines [1, 2] missing. Adding additional downstream tasks like code clone detection.**
> **A:** Thank you for the pointers, and recommending a new application for ContraCode. CodeXGLUE [2] *appears to have been made public after the ICLR deadline*, so we did not have an opportunity to benchmark on it. We are currently working on a code clone detection benchmark for JavaScript as well as porting [1] to our dataset, and will update if time permits.
>
> **Is the model pre-trained with MLM fine-tuned on the downstream task?**
> **A:** We updated Section 4 to clarify that RoBERTa is pre-trained with the MLM loss, then fine-tuned on the downstream task, the same procedure used to train ContraCode. This is also described in Section A.6.
>
>
> We’re happy to clarify more if this doesn't address your points, and again, appreciate the feedback.
>
>
> [3] Chen et al. Generative Pretraining from Pixels, 2020.
> [4] Logeswaran and Lee, An efficient framework for learning sentence representations, 2018.
> [5] Iter et al, Pretraining with Contrastive Sentence Objectives Improves Discourse Performance of Language Models, 2020.

---

> ### Author Response · Authors · 2020-11-23
> **Results for requested code clone detection benchmark**
>
> In your review, you asked us to benchmark on the code clone detection task. **We are happy to report that we added a new code clone detection benchmark in Section 4.3, Table 4 and Section A.1, Figure 5 to evaluate zero-shot transfer of self-supervised representations.**
>
> The BigCloneBench dataset used in CodeXGLUE contains Java programs, so we created a new JavaScript benchmark with student program solutions to coding interview questions from the HackerRank website. We will make this benchmark publicly available if accepted.
>
> We evaluate zero-shot code clone performance by scoring whether program A and B are clones using the cosine similarity of their respective representations. This zero-shot task is effectively analogous to the linear probe models used in PIRL, SimCLR and MoCo. It evaluates the semantic content of the learned representations without fine-tuning, which adds some useful insight, and was feasible to compute during the short rebuttal period. This is a binary classification task, so we use the area under ROC curve (AUROC) and area under precision-recall curve (average precision) metrics.
>
> Contrastive pre-training yields a significant improvement in both AUROC and AP over both a static heuristic based on textual similarity and a Transformer baseline. RoBERTa MLM does not outperform the Transformer (+0.1% AUROC, -0.4% AP). However, contrastive pre-training yields a notable improvement (+1.5% AUROC, +1.8% AP). Moreover, the hybrid loss (MLM + contrastive) yields a significant overall improvement over the Transformer (+5.27% AUROC, +5.34% AP). When compared to the textual similarity heuristic, our hybrid model achieves +10% AUROC with no fine-tuning required. Please refer to Figure 5 for the full ROC and PR curves and Table 4 for a summary of the code clone detection results.
>
> Thank you for recommending the code clone task. It’s an interesting task and a good application of contrastive learning.

---

> > ### Comment · AnonReviewer2 · 2020-11-23
> > **Interesting Results of the New Experiments**
> >
> > I'm willing to increase to my score to 6.

---

> > > ### Author Response · Authors · 2020-11-25
> > > **Thank you very much!**
> > >
> > > Thank you for increasing your score, and again for your feedback. The discussion has led to some valuable experiments that improved our paper.

---

### Official Review · AnonReviewer4 · 2020-10-31
**Review for "Contrastive Code Representation Learning"**

**Rating:** 4
**Confidence:** 4

**Review:**

This paper studies the self-supervised code functional representation learning and proposes a method called ContraCode. ContraCode utilizes some code functionality invariant transformations to generate positive pairs from the same code and negative pairs from different codes. After that, these codes pairs will be used to do the contrastive pre-training. Experiment results based on two tasks are reported.

Pros:
-	The task of  code functional representation learning is important and valuable.
-	The transformations proposed in this paper may produce some vaviance to the code while maintaining the same functionality.

Cons:
-	The superiority of the proposed method is unclear. Many self-supervised code representation learning methods are mentioned in the introduction, such as [Ben-Nun et al., 2018; Feng et al., 2020; Kanade et al., 2020]. However, this paper fail to discuss of the differences (especially the advantages) between ContraCode and other self-supervised methods empirically.
-	Since no addtional supervision is evolved, unsupervised feature leanring models are good competing baselines.. The authors are  strongly recommended to compare the performance of ContraCode with other unsupervised methods under the same training dataset (both augmented).
-	The key question is the whether the self-supervision generated by such transformation really makes any difference. Some transformation only change the formatting, which usually resulting the same feature representation because the formatting information is usually not considered in most of the feature learning methods for code. It appears that by applying the set of transformation, the code would not differ from its previous appearance much. Consequently, the feature representations generated by some unsupervised method from the original code and its transformed counterpart could be very similar to each other EVEN IF no self-supervision is enforced, which means self-supervision is not necessary.  Please clarify this be providing empirical evidences such as the portion of the changed lines or tokens from the original code, the similarity between the original code and its transformed counterpart over any two different pieces of code based on the features learned in some unsupervised way (with the same scale of training data).

---

> ### Author Response · Authors · 2020-11-16
> **We appreciate the feedback R4**
>
> Thank you for your review! We ran a suggested experiment, provide more information, and ask for a clarification below.
>
> **It appears that by applying the set of transformation, the code would not differ from its previous appearance much... Please provide empirical evidences such as the portion of the changed lines or tokens from the original code**
> **A:** Thanks for the suggestion. We added Section A.4 (“How similar are transformed programs?”) and Fig 10 with changed token statistics. We *exclude changes in whitespace*. On average, *86% of tokens* differ between negative pairs of different random programs from the dataset. In comparison, applying the 11 code transformations changes *65% of tokens* on average. The diversity of positives is close to that of negatives, which is surprising since positives implement the same functionality. Please see A.4 for more details, a histogram and more statistics.
>
> Token dissimilarity distribution after transformation (*please see A.4 for a more legible figure*):
> ```
> # Positive pairs = 1,644,353
> # Mean = 65.5; Variance = 124; SD = 11.1; Median 66.1
> # each ∎ represents a count of 15900 pairs
>
> Dissimilarity range [frequency shown in brackets]
>  0- 10 [ 0%]:
> 10- 20 [ 0%]:
> 20- 30 [ 0%]:
> 30- 40 [ 1%]:∎
> 40- 50 [ 7%]:∎∎∎∎∎∎∎
> 50- 60 [19%]:∎∎∎∎∎∎∎∎∎∎∎∎∎∎∎∎∎∎∎∎
> 60- 70 [36%]:∎∎∎∎∎∎∎∎∎∎∎∎∎∎∎∎∎∎∎∎∎∎∎∎∎∎∎∎∎∎∎∎∎∎∎∎∎
> 70- 80 [26%]:∎∎∎∎∎∎∎∎∎∎∎∎∎∎∎∎∎∎∎∎∎∎∎∎∎∎∎
> 80- 90 [ 8%]:∎∎∎∎∎∎∎∎
> 90-100 [ 0%]:
> ```
>
> With 65% token dissimilarity, we shouldn’t expect the features to be similar automatically. Yet, the t-SNE plot in Fig. 7 shows that ContraCode representations cluster together variants transformed from the same source program. We are working on more visualizations which might help clarify this.
>
> **Discuss the differences (especially the advantages) between ContraCode and other self-supervised methods empirically (Ben-Nun et al., Feng et al., Kanade et al.)**
> **A:** Ben-Nun et al. 2018 (inst2vec) isn’t a relevant baseline as their self-supervised word vector approach learns a representation of single LLVM instructions (e.g. `store i8 0, i8* %a`), not whole programs. This is limiting, as the representations don't incorporate context. Kanade et al. 2020 (cuBERT) and Feng et al. 2020 (CodeBERT) use a BERT architecture to learn to represent programs; our RoBERTa baseline uses the same architecture and loss function (masked language modeling). At the time of submission, complete data and checkpoints were not released for either paper. In our tables, RoBERTa represents our reimplementation of these BERT baseline approaches that do token reconstruction, which is also the dominant self-supervised learning strategy in NLP.
>
> **Recommended to compare the performance of ContraCode with other unsupervised methods under the same training dataset (both augmented).**
> **A:** Could you clarify what other unsupervised methods you would like to see on our dataset? We will try to run an evaluation if so, time permitting.
>
> Possible unsupervised approaches we did not benchmark include word2vec, GloVe, PCA and autoregressive LM. However, these methods are known to be inferior to masked language modeling for NLP (Devlin et al. 2019). The RoBERTa MLM baseline is expected to outperform these classic approaches. Note that it essentially trains a denoising auto-encoder using masks for corruption.
>
> Otherwise, most non-contrastive label-free methods that we cited like predicting the rotation of an image (Gidaris et al), colorization (Zhang et al.) and inpainting (Pathak et al.) were proposed in a computer vision context. These are difficult to extend to language.
>
> Thanks again for your review. Looking forward to hearing back, and please let us know if this clarifies your questions.

---

> ### Author Response · Authors · 2020-11-23
> **Update: Isolating the impact of self-supervised learning**
>
> **[Reply part 2 of 2. Please also refer to our earlier reply.]**
> We are happy to report that **we added a new benchmark to isolate the impact of self-supervised training** (MLM, ContraCode and the hybrid loss). We now include a code clone detection task in *Section 4.3, Table 4 and Figure 5*; the task predicts whether two student-written programs solve the same problem. We evaluate code clone detection by freezing the backbone representation; thus, this task is analogous to the linear probe task used to evaluate self-supervised models in computer vision e.g. Zhang et al 2016, and isolates the impact of self-supervision on the representation.
>
> Self-supervision with masked language modeling alone (as used by Kanade et al) is not enough: MLM only yields *+0.13% AUROC* and hurts AP by 0.44% over a Transformer approach. In comparison, contrastive learning alone gives *+1.47% AUROC and +1.76% AP* over a Transformer alone. Combining the objectives results in further gains, *+5.27% AUROC, +5.34% AP* over the Transformer. Over a baseline unsupervised edit distance heuristic, we achieve +10% AUROC and +8% AP.
>
> This benchmark demonstrates contrastive learning learns a higher-quality representation of program semantics and confirms the compiler-based transformations indeed capture some of the diversity of natural programs. We hope this clarifies your question regarding whether contrastive pre-training makes any difference.
>
> **[Update] Recommended to compare the performance of ContraCode with other unsupervised methods under the same training dataset (both augmented).**
> To clarify, the self-supervised masked language modeling baseline (RoBERTa) is trained on the same augmented training dataset as ContraCode. We edited A.6 to make this clearer.

---

### Author Response · Authors · 2020-11-24
**Summarizing new results, edits and discussion**

We sincerely thank all reviewers for insightful comments. The discussion has significantly improved the quality of our paper in terms of clarity, insight into our method and new results. We also thank R1 and R2 raising their scores. We summarize the key changes in our draft in one global reply.

Added code:
* We added a supplementary zip file with source code and instructions.

New experiments:
* R4’s key question is whether code transformations are diverse enough to make any difference during self-supervision. R4 requested empirical evidence such as the portion of changed tokens. We added this by **quantifying the variety of programs created by our compiler transformations**, and find our transformed pairs of programs are almost as textually distinct as random pairs of different programs. **New results: added Section A.4 and Figure 10.**
* R2 asked us to **benchmark on the code clone detection downstream task**. We created a new, appropriate dataset and added results in a zero-shot transfer setting, providing insight on the semantic content of our representations. We find RoBERTa pre-training does not improve downstream task performance. However, our method outperforms a Transformer baseline by +1.5% AUROC. Moreover, our hybrid loss with both contrastive and MLM objectives outperforms the baseline by +5.26% AUROC. **New results: added Section 4.3 and Table 4 (AUROC, AP for task), Section A.1 and Figure 5 (ROC, PR curves)**
* We **added a third t-SNE visualization to Figure 7** of representations learned by our hybrid contrastive + MLM loss. This provides insight into MLM vs contrastive learning and why they are compatible as requested by R2, and on the usefulness of self-supervision for R4.

Edits for clarity:
* Clarifications and elaboration throughout the paper, including requests by R1, R2, R4.
* Significant edits to Section 3 to clarify our proposed data augmentations through source-to-source compilers for R1. For example, we moved Algorithm 1 into the main body.
* Discussion with R2: intuition why contrastive and hybrid losses are better than MLM loss alone for programs
* Discussion with R4 on appropriate self-supervised baselines.

---

### Decision · Program_Chairs · 2021-01-07
**Final Decision**

**Decision:**

Reject

**Comment:**

This is a nice paper using contrastive learning for code representation. The idea is to generate variations on unlabeled source code (using domain knowledge) by creating equivalent version of code. Improvements over baselines on two multiple tasks are shown. While some of the reviewers liked the (and R4 should have responded), none of the reviewers found the paper exciting enough to strongly recommend its acceptance.